# Inter-agency collaboration to support the COVID-19 response and maintain malaria services in Guinea and Sierra Leone: A social network analysis

Lukas Feddern[1☉], Ibrahima Kaba[2☉], Hanna-Tina Fischer[1], Brogan Geurts[1], Habibata Balde[2], Andrea Bernasconi[1], Vitali Merhi[1], Rike Böttcher[1], Karim Dumbuya[3], Francisco Pozo-Martin[1], Thurid Bahr[1], Sara Menelik-Obbarius[1], Karoline Stein[1], Abdul Karim Mbawah[3], Alexandre Delamou[2], Heide Weishaar[1‡], Charbel El-Bcheraoui[1‡*]

1 Evidence-Based Public Health, Centre for International Health Protection, Robert Koch Institute, Berlin, Germany, 2 African Center of Excellence for the Prevention and Control of Infectious Diseases, University Gamal Abdel Nasser of Conakry, Guinea, 3 College of Medicine and Allied Health Sciences (COMAHS), University of Sierra Leone, Freetown, Sierra Leone

☉ Lukas Feddern and Ibrahima Kaba contributed equally to this work
‡ Charbel El-Bcheraoui and Heide Weishaar contributed equally to this work
* el-bcheraouic@rki.de

## Abstract

Inter-agency networks are believed to be essential to maintaining essential health services during pandemics and to strengthening the resilience of health systems. Focusing on malaria during the COVID-19 pandemic as a case study, this paper investigates the network of organisations fighting malaria in Guinea and Sierra Leone, the changes in their interactions during the pandemic, and their collaboration with organisations addressing the COVID-19 response. We used a mixed-methods approach to analyse data from 36 semi-structured interviews with respondents involved in the COVID-19 response and malaria service provision in Guinea and Sierra Leone, as well as data from a social network survey conducted with a subsample of 21 out of the 36 respondents. Results from the thematic analysis of key informant interviews were used to contextualize the social network data, which was visualized as sociograms. We found that malaria stakeholders worked closely with key stakeholders involved in containing the COVID-19 pandemic. Their ability to support the COVID-19 response and contribute to maintaining essential malaria services was facilitated by the fact that established networks of malaria stakeholders had existed in both countries which comprised a variety of national and international organisations from the public, private and civil society sector working in different areas of malaria elimination. These stakeholders were able to maintain or even intensify their collaboration during the COVID-19 pandemic. The findings suggest that in resource-limited settings, the promotion of coordination

**Data availability statement:** The data underlying this study consist of qualitative information obtained through semi-structured interviews with purposively selected mid- to senior-level representatives from national-level organizations involved in the COVID-19 response or in malaria-related activities (prevention, diagnosis, treatment, or surveillance). Due to the sensitive and identifiable nature of the data, they are not publicly available. However, data may be accessed upon reasonable request by contacting the corresponding author (CEB) or the Data Protection Officer at the Robert Koch Institute (datenschutz@rki.de), subject to institutional and ethical approvals.

**Funding:** This research project was funded through a grant provided by the Ministry of Health of the Federal Republic of Germany (Grant number: ZMI1-2521GHP914). Funding sources had no role in the study design, data collection, data analysis, writing of the manuscript, or the decision to submit it for publication. All authors had full access to all the data in the study and accept responsibility for submitting it for publication.

**Competing interests:** None of the authors have any personal, financial, professional, or intellectual bias or a conflict of interest in this project.

mechanisms among stakeholders working on specific issues can strengthen linkages and consistency, thereby contributing to overall health system resilience and pandemic response.

## Introduction

The 2014 Ebola virus disease outbreak had a devastating impact on public health and health services in West African countries [1], including Guinea and Sierra Leone [2]. The two countries were severely affected by the epidemic with 3,811 cases and 2,543 deaths in Guinea and 14,124 cases and 3,956 deaths in Sierra Leone, respectively [3]. The negative consequences comprised a significant increase in non-Ebola morbidity and mortality, particularly from malaria, a leading cause of death in both countries [4]. The adverse impact on public health and health services was, in part, due to weaknesses in the respective health systems and a lack of adequate leadership and coordination [5–7]. The Ebola outbreak exposed the chronic underfunding of these health systems and highlighted the insufficient domestic investment in primary health care [8]. Following the outbreak, donor commitment shifted from immediate response efforts to rebuilding and strengthening foundations of the healthcare systems in Ebola-affected countries [9]. This shift included ensuring funding to support programs disrupted during the Ebola crisis, such as malaria, to provide continuous essential services [10,11]. Four years later, at the onset of the study in 2021, Guinea reported approximately 4.4 million cases of malaria, whereas Sierra Leone recorded 2.9 million cases. During the same year, Guinea documented 7.2 million cases of COVID-19 (an increase from 4.7 million in 2020), while Sierra Leone reported 921,000 cases (a decrease from 2.6 million cases in the previous year) [4]. Recent studies suggest that the health systems in Guinea and Sierra Leone were better able to maintain essential health services during COVID-19 than during Ebola [12] and found ways to absorb the shock, adapt to the challenges and transform to improve responsiveness [13].

Against the backdrop of this history of public health crises and funding efforts to strengthen health systems, Guinea and Sierra Leone provide an ideal setting for a natural experiment to assess collaboration and coordination and its role in maintaining essential health services during a public health crisis. This study investigates how malaria stakeholders in Guinea and Sierra Leone collaborated during the COVID-19 pandemic to maintain malaria services. Additionally, it aims to determine whether, and how, these stakeholders collaborated with organizations involved in the pandemic response. We chose malaria as a case study because it is one of the leading causes of death in both countries (accounting for 9.9% and 16.17% of all deaths in Guinea and Sierra Leone, respectively) [4], and because its effective elimination requires a combination of preventive activities (e.g., the distribution of insecticide-treated bed nets, vector control, chemo-prophylaxis in pregnancy), as well as testing, treatment and surveillance interventions.

## Materials and methods

We adopted a mixed-method social network approach comprising two components: key informant interviews to gain in-depth insights into stakeholders' interactions, and a social network survey to map the collaboration between these stakeholders involved in malaria services and their engagement in COVID-19.

### Literature review

Prior to primary data collection, we conducted a preliminary search of peer-reviewed scientific literature and a desk review of relevant national documents, both pre-existing and developed during the response to the COVID-19 pandemic, to understand the malaria services and COVID-19 response in the two countries. Among others, we searched the electronic databases PubMed, EMBASE, Bielefeld Academic Search Engine (BASE), and WHO's Institutional Repository for Information Sharing (IRIS), the search engine Google Scholar, as well as the websites of key organisations working on malaria and COVID-19 in Guinea and Sierra Leone. We also carried out a fact-finding mission to a number of urban and rural sites in Guinea and Sierra Leone. The literature search, desk review and fact-finding mission jointly served to compile a list of key malaria and COVID-19 stakeholders in Guinea and Sierra Leone and gather contextual data to refine the data collection tools. The preliminary list was augmented by national experts working on the thematic issues in both countries with information on the pre-identified stakeholders as well as with information on any additional stakeholders perceived as relevant in the national context. In our study, we considered stakeholders as any organisation mentioned in the literature and/or public documents as working on the prevention, diagnosis, treatment or surveillance of malaria or on the COVID-19 response at the national level. The final stakeholder list was then used to select interviewees and compile a roster of organisations for the social network survey.

### Key informant interviews

We conducted semi-structured interviews with purposively sampled mid- to senior-level representatives from national-level organizations involved in either the COVID-19 response or in the prevention, diagnosis, treatment or surveillance of malaria. Respondents were recruited to the study from 11 March to 29 March 2022 in Guinea, and from 5 March to 26 March 2022 in Sierra Leone. In Guinea, we interviewed 16 organizations. In most interviews, only one person responded to our questions. However, in one interview, an additional individual from the same institution was present and contributed their views to some of the questions. In Sierra Leone, 19 organizations were interviewed, with one respondent from each organization. One team member took notes throughout the interviews. The questionnaire pertaining to the social network survey was programmed on ODK, an open-source data collection software, and answers were collected on tablets by the interviewers. Respondents included representatives from both Ministries of Health, non-governmental organisations, international organisations, and bilateral and multilateral donors in both countries. The aim of the interviews was to explore 1) the effect of the pandemic on health service delivery in Guinea and Sierra Leone, 2) the ways in which the health systems in the two countries responded during the COVID-19 pandemic, and 3) the collaboration between different organisations. In Guinea, all interviews were conducted in French (except for one interview which was conducted in English). In Sierra Leone, all interviews were conducted in English. All interviews were audio recorded and on average lasted 43 minutes (min = 24 minutes; max = 103 minutes). Written consent was obtained prior to the interviews.

The qualitative data was transcribed in its original language, and translated into English, before it was coded, and analysed using a framework approach to thematic analysis. A codebook was developed using pre-defined codes that came from the interview topic guides and in vivo codes that were identified from wording used by respondents [14]. The development of the codebook was an interactive, collaborative process that involved piloting the draft codebook on select transcripts to refine the codes until consensus was reached among the research team. All transcripts were independently coded using NVivo® qualitative data management software version 12 by at least two coders. Tests of interrater reliability were conducted to ensure consistency across the coding team.

## Social network survey

We developed a questionnaire to map the structure of, and interactions within, the network of organizations involved in the COVID-19 response and malaria service provision in Guinea and Sierra Leone. We asked respondents to provide information about the network members (i.e., the respondent's organization and any organizations with which their organization collaborated), network members' attributes (i.e., the characteristics of collaborating organisations), and relational data (i.e., the interactions the respondent maintained with other network members) (Table 1). Respondents were asked to assess interactions and to recall changes in the frequency and quality of interactions since the beginning of the COVID-19 pandemic. The survey was conducted among a subsample of the respondents who participated in a key informant interview, with only the most senior representative of each organisation or department being selected.

Based on publicly available information, we grouped the network members into seven organisational categories: bilateral partners (organisations headquartered in a high income country, partnering directly with an organisation in Guinea or Sierra Leone, respectively), international non-governmental organizations (NGOs), international organizations, multilateral partners (organisations headquartered in a high-income country and receiving contributions from various sources, directly partnering with an organisation in Guinea or Sierra Leone, respectively), national government, national NGOs, private (for-profit) organizations, and others. To analyse the structure of the networks, we calculated network measures, including network composition, degree centrality (a measure of how many other network members an individual network member is connected to) [14], density (a measure of how many relationships between network members exist compared to all relationships potentially possible) [15] and centralisation (a measure to assess to what extent a network is dominated by one or a few central network members) [16]. Subsequently, we created sociogrammes to illustrate the structure of the networks and the interactions between the stakeholders in the two countries. All network statistics and visualizations were done using the R Statistical Software [17], specifically the tidyverse, data.table, ggraph, and tidygraph packages.

The study received ethical approval from the Ethics Committee of the Ärztekammer Berlin in Germany (Eth-76/21), the Ethics and Scientific Review Committee of Sierra Leone (March 15, 2022), and the National Health Research Ethics Committee in Guinea (013/CNERS/21). The documents submitted for ethical approval contained, among others, the study information sheet, the informed consent form, the interview topic guide and the social network survey.

## Results

Our findings are presented in alignment with the three main objectives of this study. First, we describe the collaboration and coordination between organisations working on malaria and those involved in the COVID-19 response. Next, we focus on the network of stakeholders working on malaria and their reports of collaboration before presenting findings on changes in interactions that were reported between malaria stakeholders since the start of the COVID-19 pandemic.

Table 1. Questionnaire items for the social network survey.

| Type of data | Information |
| --- | --- |
| Nodes: Network members | • Respondent's organization<br>• Roster of organisations the respondent's organisation collaborates with<br>• Additional organisations that the respondent's organisation collaborates with |
| Attributes: Characteristics of network members | • Focus of organization's work in terms of malaria and/or COVID-19: malaria prevention, malaria diagnosis, malaria treatment, malaria surveillance, COVID-19 response |
| Relational data: Interactions with network members (only collected for malaria members) | • Frequency of interactions: daily, weekly, monthly or less frequently<br>• Changes in the frequency of interaction since the start of the COVID-19 pandemic: increased, no change, decreased<br>• Changes in the quality of interactions since the start of the COVID-19 pandemic: improved, no change, deteriorated |

## Collaboration and coordination *between* malaria and COVID-19 stakeholders

Respondents in both countries reported that since the start of the COVID-19 pandemic, malaria stakeholders had closely collaborated with organisations involved in the COVID-19 response. This close collaboration was evident in the structural network analysis. The networks that were generated based on the malaria stakeholders' accounts of collaboration with organisations working on COVID-19 comprised 26 and 38 organisations in Guinea and Sierra Leone, respectively (Fig 1). The stakeholders with the highest degree centrality scores in the Guinean network were the Programme National de Lutte Contre le Paludisme (National Program for the Fight Against Malaria, degree centrality: 41) and the Catholic Relief Services (degree centrality: 32). In Sierra Leone, organisations with the highest degree centrality scores included the Catholic Relief Services (degree centrality: 42) and the Global Fund to Fight AIDS, Tuberculosis and Malaria (degree centrality: 39).

The interview data complemented the findings from the structural network analysis. Respondents recalled that «*everybody was just concentrating on fighting COVID, because that was the issue at the point in time*» (Sierra Leone, National NGO). They explained how malaria stakeholders had aligned their activities to support the implementation of the COVID-19 response. This included adapting malaria-focused social and behavioural change communication activities, such as radio programs to include information on COVID-19 related information. It also included leveraging existing community outreach structures to disseminate information about COVID-19. A respondent from Sierra Leone recalled efforts to harmonise COVID- and malaria-related messages:

> «*What we did first was to try to […] harmonise the message COVID to that of malaria so that we can improve health seeking behaviour. […] For instance, when [NaCOVERC, the National COVID-19 Emergency Response Center] were printing the banner, […] we incorporated our own key message just underneath the COVID-19 message. So if you see the banners that were out there, […] both COVID and malaria messages were going along side by side.*» (Sierra Leone, Bilateral Partner)

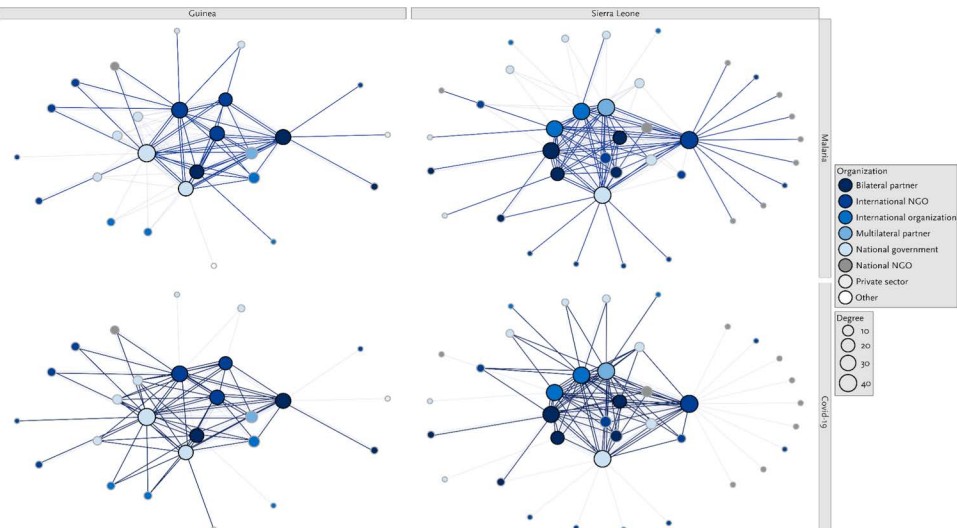

**Fig 1. Sociogramme of malaria- and COVID-19-related interactions by organization type in Guinea and Sierra Leone.** Legends: The figures on top show the interactions related to malaria. These malaria-related interactions are represented by the bolded lines. The figures on the bottom show the interactions related to COVID-19 between the different stakeholders. These COVID-19-related interactions are represented by the bolded lines.

The interviews revealed that different approaches had been taken to coordinating the COVID-19 response in the two countries. Both countries had pursued a Whole-of-Government approach, meaning the response had been initiated via a presidential decree and had involved all ministries. Respondents reported that in Guinea, the Agence Nationale de Sécurité Sanitaire (National Agency of Health Security, ANSS) had been coordinating the response. The equivalent authority in Sierra Leone, the Department of Health Security and Emergencies, had not, reportedly, played an equally prominent role, but instead, a temporary body called NaCOVERC had been established. With regard to this approach, the concern was voiced that, due to NaCOVERC being a temporary structure, lessons learned during the pandemic would disappear once the response centre was dissolved. Informants stated that the same happened during and after Ebola; the National Ebola Response Center (NERC) had been established as a central coordinating body but then dissolved once the epidemic was over. A Sierra Leonean respondent criticised the massive financial investment that had been made to establish NaCOVERC, contrasting it with the budget that was available to the Ministry of Health:

«*The amount of money spent on NaCOVERC alone is more than… Even one tenth of that is not given to the entire Ministry of Health for other programs because the government focus is to control the pandemic. […] So the attention was on COVID and […] government pay more money to NaCOVERC activity than to the Ministry of Health up till now.*»
(Sierra Leone, National Government)

**Structure, function, nature and composition of the malaria networks**

Focusing on the malaria networks in both Guinea and Sierra Leone, the structural network analysis showed the diversity of stakeholders involved (Fig 2). The Guinean network consisted of seven members working on malaria, including international NGOs (n = 3), bilateral partners (n = 2), and national government organisations (n = 2). In Sierra Leone, the network consisted of a total of eight members working in malaria, including bilateral partners (n = 3), international organizations (n = 2), and one international NGO, multilateral partner and national government, each.

Both malaria networks had high density scores (Guinea: 0.95, Sierra Leone: 1.00), with almost all respondents reporting collaboration with all other members in the network. Correspondingly, centralisation was low (Guinea: 1, Sierra Leone: 1), and the diameters for both networks were very short (Guinea: 2, Sierra Leone: 1), indicating low hierarchy, equitable interactions, and multiple opportunities for direct collaboration and malaria-related communication. An analysis of the frequency of interactions revealed that the malaria network members in Guinea engaged in slightly more frequent interaction than the malaria network members in Sierra Leone (Fig 2).

The key informant interviews confirmed the findings from the network survey, as respondents reported that they had well-established, good collaboration with the different stakeholders working on malaria in both countries. Guinean as well as Sierra Leonean respondents mentioned that the government, notably the Ministry of Health and Public Hygiene in Guinea and the Ministry of Health and Sanitation in Sierra Leone, were key players in the fight against malaria, and that the government-initiated National Malaria Control Program (NMCP) was a key initiative that drove malaria elimination. As the following quote by a respondent from Sierra Leone highlights, other stakeholders frequently saw their own activities as supporting the NMCP:

«*Our role is to support the ministry of health, [...] we support the National Malaria Control Program.*» (Sierra Leone, International NGO)

In both countries, the international NGO Catholic Relief Services was reported to be the main recipient and co-principal investigator alongside the Ministry of Health of the Global Fund to Fight AIDS, Tuberculosis and Malaria grant, and thus the main technical coordinator of malaria activities in-country. The «*dialogue process*» (Sierra Leone, Multilateral Partner) and the close collaboration between sub-recipients of the grant that characterised the collaboration were positively stressed.

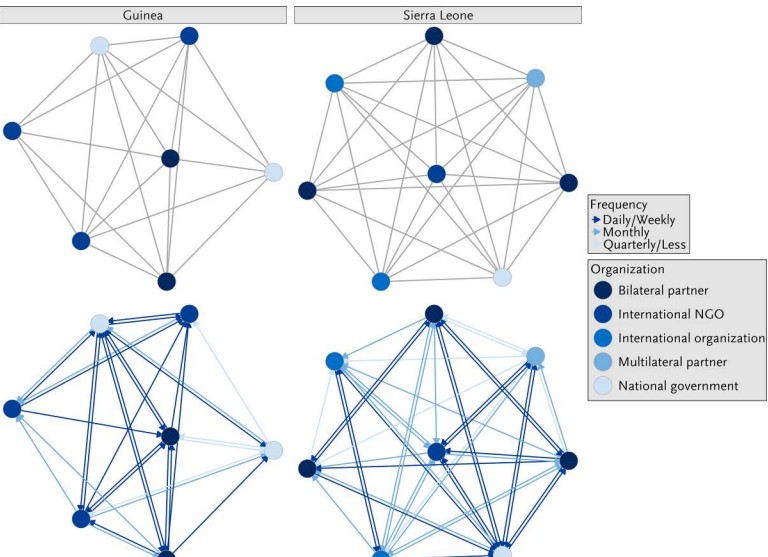

**Fig 2. Sociogramme of malaria-related interactions between health system stakeholders presented by organization type in Guinea and Sierra Leone.** Legends: The figures on the bottom display the reported frequency and direction of interactions.

In both countries, the structural network analysis showed that malaria stakeholders jointly provided a broad spectrum of malaria services, including prevention, diagnosis, treatment, and surveillance. In Guinea, five of the seven network members provided all four malaria services, one member provided prevention, treatment, and surveillance services, and one member worked in prevention and surveillance. In Sierra Leone, four of the eight network members provided all four malaria services, one member worked in diagnosis and treatment, one was involved in both prevention and surveillance, and one each worked in only surveillance and only prevention. Prevention and surveillance were the services that were most prominently covered in both countries, with the Guinean and the Sierra Leonean network comprising six members each providing these services. Treatment services were provided by five members in both countries, and diagnostic services were provided by four members in Guinea and five members in Sierra Leone.

### Changes in interactions between malaria stakeholders during the COVID-19 pandemic

The analysis showed that collaboration between malaria stakeholders was maintained and sometimes even improved and intensified throughout the COVID-19 pandemic (Fig 3). The majority of respondents reported that the quality of the interactions with other malaria stakeholders had remained the same, and some even reported that it had improved over the course of the pandemic. A slightly stronger improvement in the quality of interactions across the network was reported in Sierra Leone compared to Guinea. Most respondents from both Guinea and Sierra Leone further recalled that the frequency of interactions between malaria stakeholders had remained unchanged or had increased. Three network members in Guinea and six network members in Sierra Leone reported decreased frequency in interaction since the start of the pandemic.

The interviews confirmed the findings from the structural network analysis. Respondents highlighted that they had been keen to maintain good collaboration between malaria stakeholders during the pandemic in order to ensure the continuity of services. In Sierra Leone, the need to overcome public mistrust in the health system was mentioned as a key driver for ensuring continuous collaboration. Respondents from Guinea reported that health services had been interrupted at the start of the COVID-19 pandemic due to contact restrictions. They recalled that given, the potential detrimental

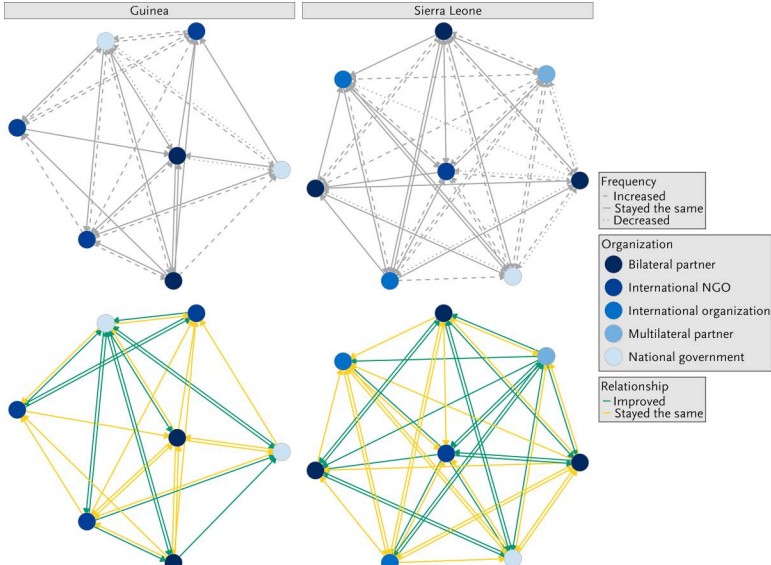

**Fig 3. Network mapping of health system stakeholders and changes in the frequency and quality of their interactions.** Legends: This is relative to the beginning of the COVID-19 pandemic by organization type in Guinea and Sierra Leone in 2022.

consequences of these interruptions, malaria stakeholders, in discussion with other government officials, agreed to resume service provision under certain hygiene restrictions:

> «*We began to feel the effects of COVID on the different programs, especially on malaria. […] After we felt the impact, we came to the Ministry to plead that we must fight against COVID, but we must let some programs adapt their activities, otherwise there will be several epidemics. Because the Expanded Program on Immunization could not vaccinate, the malaria could not fight against malaria. So, the authorities have understood, they have let it go, but taking into account the barrier measures.*» (Guinea, National Government)

Regular meetings were highlighted as important to allow stakeholders to join forces, coordinate support, and ensure the continuity of services. Respondents highlighted that face-to-face meetings had been replaced by online meetings and other forms of interaction. Only a small number of stakeholders complained that their expertise had not been used during COVID-19. The majority, however, provided accounts of the good collaboration that had continued to exist between malaria stakeholders.

## Discussion

This paper presents findings from a mixed-methods network study investigating the collaboration of organisations working on malaria and organisations involved in the COVID-19 response during the pandemic. It assesses the collaborative efforts of these stakeholders in responding to COVID-19 and maintaining essential health services during the pandemic in Guinea and Sierra Leone. Our analysis shows that in both countries, established networks existed between organisations working on malaria prior to the pandemic. Throughout the pandemic, these networks continued to exist and were leveraged to support the pandemic response and ensure the maintenance of essential malaria services. Our study provides evidence of the extent and nature of collaboration between organisations working to maintain health services during the COVID-19 pandemic in Guinea and Sierra Leone, two low-income countries significantly impacted by the Ebola epidemic

in West Africa. It also shows that established issue-specific networks and related collaborations can be leveraged to strengthen a pandemic response, contribute to the maintenance of services, and consequently to the resilience of the health system.

Our analysis shows that in Guinea and Sierra Leone, networks exist which provide a comprehensive set of services to fight malaria. Covering the full spectrum of malaria services, ranging from prevention to diagnosis, treatment, and surveillance, these networks could be maintained during a health system crisis. In both countries, members of the networks represent various sectors of society, including government, civil society, and international donors. Our analysis also highlights that the networks in both countries are guided by a joint goal and agreed strategy and that they are "living" and active, evidenced by the regular exchanges and interactions that exist between all network members.

Our analysis of the networks' performance during the COVID-19 pandemic shows that the malaria networks in both countries were leveraged in responding to the pandemic. For example, they were helpful in disseminating information about COVID-19 and reaching local communities. Arguably even more importantly, and as reported by respondents who have first-hand knowledge of the response in Guinea and Sierra Leone, the maintenance of networks contributed to the continuity of essential health services throughout the pandemic, including services relating to malaria prevention, diagnosis, surveillance, and treatment. Also, despite restrictions deriving from COVID-related public health and social measures, malaria stakeholders not only continued frequent, high-quality collaboration among themselves, but also connected with a larger number of organisations involved in the pandemic response. The analysis suggests that the well-established network served as a solid foundation and a facilitator for the implementation of the pandemic response. Established structures of communication and collaboration appear to have been helpful in rolling out the pandemic response at scale and incorporating response activities within the health system. Our findings indicate that networks around and within the health system play a pivotal role in response coordination, as well as in sustaining essential health services during crises. Networks facilitate rapid information exchange, resource sharing, and collaborative decision-making, which may mitigate service disruptions. Hence, decision makers should consider strengthening such networks prior to public health emergencies to ensure continuity of health services during crises. Future investments should consider network development as part of emergency preparedness strategies.

One of the reasons why organisations working on malaria were well-connected and able to support the pandemic response by employing their communication and collaboration structures might be that the collaboration between these stakeholders had been fostered and promoted prior to as well as throughout the pandemic. In fact, the improvement of the coordination between partners involved in malaria service provision in Guinea and Sierra Leone is the designated aim of the funding provided by the Global Fund to Fight AIDS, Tuberculosis and Malaria Country Coordinating Mechanism [18]. Through this mechanism, the Global Fund to Fight AIDS, Tuberculosis and Malaria requires government and non-government stakeholders to coordinate their efforts and establish public-private partnerships to implement national programmes to combat specific health issues, including malaria [18]. Our analysis provides some indication that it is crucial to acknowledge the importance of collaboration and coordination for health system performance and for ensuring essential health services in low- and middle-income countries. In fact, previous research has shown that cooperation, collaboration and leadership were important aspects in countering both Ebola and COVID-19 in Sierra Leone [19]. Given that resources for health in low- and middle-income countries are scarce, coordinating efforts is of paramount importance. Respective funding to foster coordination and collaboration seems to be able to facilitate and improve not only illness-specific services but also service provision across the public health sector more broadly. The targeted funding of partnerships might thus be a promising path to achieve two aims at the same time: the strengthening of linkages and consistency on specific health issues, and overall health system strengthening.

The COVID-19 pandemic has highlighted the vulnerability of essential services for other infectious diseases, such as tuberculosis, with disruptions reported in several high-income countries [20]. In contrast, our study found that

malaria-related services were largely maintained during the pandemic in Guinea and Sierra Leone. This highlights the role of established networks in maintaining essential health services during crises. Future pandemic preparedness plans, especially in low-income countries, should explicitly integrate the development of networks around the health system into strategies aimed at sustaining services for tuberculosis, malaria, and other infectious diseases, alongside emergency response efforts.

Our study has a number of limitations. First, due to resource constraints and the substantial efforts that empirical social network research requires, we only obtained social network data from representatives of organisations active in malaria elimination at national level. We did not collect social network data from stakeholders working at regional or communal levels and not from their collaborating partners who worked in other fields, including on COVID-19. This means that we were unable to verify the information provided by national-level malaria stakeholders, and that the network that we describe reflects the perspective of this specific group of stakeholders. Second, the selection of key informants and social network survey respondents may be limited by us missing relevant stakeholders. As outlined in detail above, we conducted comprehensive searches of the available literature and documents and consulted experts with knowledge of the local context to identify all relevant individuals and organisations. We further gave social network survey respondents the option of naming additional organisations they collaborated with which had not initially been included in the network roster. However, a risk remains that not all relevant stakeholders were included in the study and that our sample might therefore be biased. Third, given that the terms «interaction», «relationship», and «collaboration», were not specified when collecting data for the social network survey, respondents might have interpreted these terms in different ways. To alleviate this limitation and to understand the nature of collaboration better, we elicited in-depth information via the key informant interviews. Fourth, our study is cross-sectional, i.e., information about changes in the frequency and quality of interactions might be influenced by recall bias, particularly as interviews were conducted in March 2022 when the pandemic was considered to have largely been over in both countries. Future longitudinal research could explore networks over time to gather empirical evidence about actual changes in collaboration.

## Conclusions

This mixed-method study emphasises the importance of collaboration and networks in global health and highlights that collaboration between stakeholders working on specific health issues can not only ensure consistency and alignment of activities in non-emergency times, but is also of crucial importance during public health crises. Efforts to foster collaborative networks on key public health issues have considerable potential to strengthen health systems and improve health system resilience.

## Acknowledgments

We would like to express our gratitude to the respondents who took the time to participate in our interviews and contribute their knowledge and expertise to this study. We also want to acknowledge the support we received from Sameh Al-Awlaqi (RKI) in Germany, Fodé Youssouf Camara (CEA-PCMT) in Guinea, Alioune Camara (NMCP) in Guinea, and Bockarie Kemokai (COMAHS) in Sierra Leone in administering the research project.

## Author contributions

**Conceptualization:** Lukas Feddern, Ibrahima Kaba, Habibata Balde, Heide Weishaar, Charbel El-Bcheraoui.

**Data curation:** Lukas Feddern, Ibrahima Kaba, Karim Dumbuya, Abdul Karim Mbawah.

**Formal analysis:** Lukas Feddern.

**Funding acquisition:** Heide Weishaar, Charbel El-Bcheraoui.

**Investigation:** Lukas Feddern, Ibrahima Kaba, Habibata Balde, Karim Dumbuya, Abdul Karim Mbawah.

**Methodology:** Lukas Feddern, Heide Weishaar.

**Project administration:** Ibrahima Kaba, Habibata Balde, Vitali Merhi, Karim Dumbuya, Sara Menelik-Obbarius, Karoline Stein.

**Resources:** Lukas Feddern, Heide Weishaar, Charbel El-Bcheraoui.

**Software:** Lukas Feddern.

**Supervision:** Heide Weishaar, Charbel El-Bcheraoui.

**Validation:** Alexandre Delamou, Heide Weishaar, Charbel El-Bcheraoui.

**Visualization:** Lukas Feddern.

**Writing – original draft:** Lukas Feddern, Ibrahima Kaba.

**Writing – review & editing:** Lukas Feddern, Ibrahima Kaba, Hanna-Tina Fischer, Brogan Geurts, Habibata Balde, Andrea Bernasconi, Rike Böttcher, Karim Dumbuya, Francisco Pozo-Martin, Thurid Bahr, Sara Menelik-Obbarius, Karoline Stein, Abdul Karim Mbawah, Alexandre Delamou, Heide Weishaar, Charbel El-Bcheraoui.

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
