## [Decision Letter · Decision Letter 0]

PONE-D-24-35950Inter-agency collaboration to support the COVID-19 response and maintain malaria services in Guinea and Sierra Leone: A social network analysisPLOS ONE

Dear Dr. El Bcheraoui, Thank you for submitting your manuscript to PLOS ONE. After careful consideration, we feel that it has merit but does not fully meet PLOS ONE’s publication criteria as it currently stands. Therefore, we invite you to submit a revised version of the manuscript that addresses the points raised during the review process.

We look forward to receiving your revised manuscript.

Kind regards,

Simone Lanini

Academic Editor

PLOS ONE

Journal Requirements:

4. In the online submission form you indicate that your data is not available for proprietary reasons and have provided a contact point for accessing this data. Please note that your current contact point is a co-author on this manuscript. According to our Data Policy, the contact point must not be an author on the manuscript and must be an institutional contact, ideally not an individual. Please revise your data statement to a non-author institutional point of contact, such as a data access or ethics committee, and send this to us via return email. Please also include contact information for the third party organization, and please include the full citation of where the data can be found.

“This research project was funded through a grant provided by the Ministry of Health of the Federal Republic of Germany (Grant number: ZMI1-2521GHP914). Funding sources had no role in the study design, data collection, data analysis, writing of the manuscript, or the decision to submit it for publication.”

“This research project was funded through a grant provided by the Ministry of Health of the Federal Republic of Germany (Grant number: ZMI1-2521GHP914). Funding sources had no role in the study design, data collection, data analysis, writing of the manuscript, or the decision to submit it for publication. All authors had full access to all the data in the study and accept responsibility to submit it for publication.”

7. Please remove your figures from within your manuscript file, leaving only the individual TIFF/EPS image files, uploaded separately. These will be automatically included in the reviewers’ PDF

Additional Editor Comments (if provided):

your manuscript has been reviewed by me and two independent reviewers. The reviewers appreciate the quality of your work, noting that it is well-conceived, well-written, and addresses an important research question. 

Given the significance of your study on inter-agency networks and their role in sustaining essential health services during pandemics, the reviewer recommends minor revisions before the manuscript can be considered for publication.

We kindly ask you to revise the manuscript accordingly and submit a revised version along with a point-by-point response to the reviewer’s comments. Please highlight any changes made in the revised manuscript for easier reference. 

Reviewers' comments:

Reviewer's Responses to Questions

**Comments to the Author**

1. Is the manuscript technically sound, and do the data support the conclusions?

Reviewer #1: Yes

Reviewer #2: Yes

2. Has the statistical analysis been performed appropriately and rigorously? 

Reviewer #1: Yes

Reviewer #2: Yes

3. Have the authors made all data underlying the findings in their manuscript fully available?

Reviewer #1: Yes

Reviewer #2: Yes

4. Is the manuscript presented in an intelligible fashion and written in standard English?

Reviewer #1: Yes

Reviewer #2: Yes

5. Review Comments to the Author

Reviewer #1: This is a well written manuscript and was easy to read with a nice flow. I have a few observations and questions:

Line 126: Authors used participants and then key informants and respondents. For uniformity authors should choose one word for example use respondents all through or key informants and respondents were appropriate.

Line 137: Which language were the interviews and survey conducted? Earlier in the text authors mentioned English as the language used to interview then this line says translated to English. If this is truly a translation, authors should provide additional details on how this was done and who conducted the interviews and in which language.

Line 147:Authors to please provide the number of organizations involved and how many respondents participated per organization and per country. How were these questionnaires administered- self, electronically, etc?

I was not so excited about the GSL acronym, is this a standardized acronym?

Reviewer #2: In this paper, the authors examine the role of inter-agency networks in sustaining essential health services during pandemics. Using malaria control efforts during the COVID-19 pandemic as a case study, the paper explores the network of organizations combating malaria in Guinea and Sierra Leone, the shifts in their interactions during the pandemic, and their collaboration with organizations involved in the COVID-19 response. The paper is well-conceived, well-written, and presents an interesting question.

In my opinion, it would be interesting for the reader if the introduction included an overview of both the number of COVID-19 notifications (if available) and malaria incidence during the study period in the countries.

Moreover, the discussion could be strengthened by further refining suggestions on the role of pandemic networks in maintaining adequate healthcare services for infectious diseases. A key lesson from the recent COVID-19 pandemic is that many infectious diseases, such as tuberculosis ( García-García JM, COVID-19 Hampered Diagnosis of TB Infection in France, Italy, Spain and the United Kingdom. Arch Bronconeumol. 2022 Nov;58(11):783-785) were significantly impacted. It is crucial for scientific papers to contribute to addressing this issue, particularly in low-income countries, by emphasizing the need to incorporate these specific activities into future pandemic preparedness plans.

6. PLOS authors have the option to publish the peer review history of their article (what does this mean? ). If published, this will include your full peer review and any attached files.

**Do you want your identity to be public for this peer review?** For information about this choice, including consent withdrawal, please see our Privacy Policy .

Reviewer #1: **Yes: ** Sophia Osawe

Reviewer #2: No

---

## [Author Response · Author response to Decision Letter 1]

6 May 2025

Please find the response to reviewers attached as a word document

---

## [Editor Report · Decision Letter 1]

Inter-agency collaboration to support the COVID-19 response and maintain malaria services in Guinea and Sierra Leone: A social network analysis

PONE-D-24-35950R1

Dear Dr. El Bcheraoui,

We’re pleased to inform you that your manuscript has been judged scientifically suitable for publication and will be formally accepted for publication once it meets all outstanding technical requirements.

Kind regards,

Simone Lanini

Academic Editor

PLOS ONE
---

## [Editor Report · Acceptance letter]

PONE-D-24-35950R1

PLOS ONE

Dear Dr. El-Bcheraoui,

I'm pleased to inform you that your manuscript has been deemed suitable for publication in PLOS ONE. Congratulations! Your manuscript is now being handed over to our production team.

Kind regards,

on behalf of

Prof. Simone Lanini

Academic Editor

PLOS ONE